# Molecular Epidemiology Surveillance of SARS-CoV-2: Mutations and Genetic Diversity One Year after Emerging

**DOI:** 10.3390/pathogens10020184

**Published:** 2021-02-09

**Authors:** Alejandro Flores-Alanis, Armando Cruz-Rangel, Flor Rodríguez-Gómez, James González, Carlos Alberto Torres-Guerrero, Gabriela Delgado, Alejandro Cravioto, Rosario Morales-Espinosa

**Affiliations:** 1Departamento de Microbiología y Parasitología, Facultad de Medicina, Universidad Nacional Autónoma de México, Mexico City 04360, Mexico; bioalejandrofa@gmail.com (A.F.-A.); delgados@unam.mx (G.D.); dracravioto@hotmail.com (A.C.); 2Laboratorio de Bioquímica de Enfermedades Crónicas, Instituto Nacional de Medicina Genómica, Mexico City 14610, Mexico; acruz@inmegen.gob.mx; 3Departamento de Ciencias Computacionales, Centro Universitario de Ciencias Exactas e Ingenierías, Universidad de Guadalajara, Guadalajara 44430, Jalisco, Mexico; fiores.flor@gmail.com; 4Departamento de Biología Celular, Facultad de Ciencias, Universidad Nacional Autónoma de México, Mexico City 04510, Mexico; james@ciencias.unam.mx; 5Posgrado en Edafología, Colegio de Postgraduados, Mexico City 56230, Mexico; cartogue86@gmail.com

**Keywords:** SARS-CoV-2, genetic diversity, molecular surveillance, natural selection, non-synonymous substitution

## Abstract

In December 2019, the first cases of the novel severe acute respiratory syndrome coronavirus 2 (SARS-CoV-2) were identified in the city of Wuhan, China. Since then, it has spread worldwide with new mutations being reported. The aim of the present study was to monitor the changes in genetic diversity and track non-synonymous substitutions (*dN*) that could be implicated in the fitness of SARS-CoV-2 and its spread in different regions between December 2019 and November 2020. We analyzed 2213 complete genomes from six geographical regions worldwide, which were downloaded from GenBank and GISAID databases. Although SARS-CoV-2 presented low genetic diversity, there has been an increase over time, with the presence of several hotspot mutations throughout its genome. We identified seven frequent mutations that resulted in *dN* substitutions. Two of them, C14408T>P323L and A23403G>D614G, located in the nsp12 and Spike protein, respectively, emerged early in the pandemic and showed a considerable increase in frequency over time. Two other mutations, A1163T>I120F in nsp2 and G22992A>S477N in the Spike protein, emerged recently and have spread in Oceania and Europe. There were associations of P323L, D614G, R203K and G204R substitutions with disease severity. Continuous molecular surveillance of SARS-CoV-2 will be necessary to detect and describe the transmission dynamics of new variants of the virus with clinical relevance. This information is important to improve programs to control the virus.

## 1. Introduction

Following reports of a new infectious disease in the city of Wuhan, China, in December 2019, the subsequent global pandemic has led to 82,579,768 confirmed cases and 1,818,849 deaths up to 2 January 2021 [1]. The infectious agent responsible for this pandemic was identified as a virus of the Coronavirus family (CoVs), which was named as severe acute respiratory syndrome CoV 2 (SARS-CoV-2) [2], while the disease caused by this virus was called COVID-19 (Coronavirus disease 2019). SARS-CoV-2 is a positive, simple-strand RNA virus with a genome of approximately 29 Kb in size that is organized into 11 open reading frames (ORFs) [3]. The first ORF represents approximately 70% of the viral genome, which is composed of two overlapping ORFs (ORF1a and ORF1b). These ORFs encode two polypeptides that are processed into 16 non-structural proteins (nsp1-16). The main non-structural proteins include RNA-dependent RNA polymerase (RdRp or nsp12) and a 3’ > 5’ exonuclease (ExonN or nsp14) [4]. The remaining ORFs encode the following four structural proteins: the Spike surface glycoprotein (S), an envelope protein (E), a membrane protein (M), the nucleocapsid protein (N) and other accessory proteins (ORF3a, ORF6, ORF7a/b, ORF8 and ORF10) [5,6]. 

An important factor in the evolution of RNA viruses is their high mutation rate (10^−6^ to 10^–4^ substitutions/nucleotide/cell infection) [7]. This phenomenon can be explained partially because the RNA polymerase cannot correct mistakes during genome replication [8]. However, CoVs possess an ExonN with the capacity to correct mistakes that occur during replication [9]. This feature has contributed to the low mutation rate of CoVs compared to other RNA viruses [10,11,12]. 

When a virus is well adapted to its environment, the establishment of new mutations in the virus population is not favored because most mutations become deleterious (purifying selection). In general, mutations that increase in frequency could be advantageous and fixed by positive selection, or they could be neutral and fixed by genetic drift. It is important to note that when neutral mutations increase in frequency, they can be confused with positive natural selection [8,13]. Therefore, in studies involving the evolutionary dynamics of a new pathogenic virus, such as SARS-CoV-2, it is important to know if the increase in the frequency of mutations is due to natural selection, in order to determine the possible consequences for its fitness, such as increased infectiousness and pathogenicity, or due to adaptation, thereby becoming drug resistant or having the ability to evade the immune system. 

The aim of the present study was to monitor the genetic diversity of SARS-CoV-2 and use molecular epidemiology to track non-synonymous substitutions (*dN*) that could be implicated in the fitness of SARS-CoV-2 and its spread in different regions between December 2019 and November 2020. The information generated will be useful to understand the evolutionary dynamics of SARS-CoV-2 better in order to improve intervention measures against it.

## 2. Results

### 2.1. Global Genetic Diversity of SARS-CoV-2

A comparison among the 2213 SARS-CoV-2 genomes showed high nucleotide identity (99.9–100%), with an average pairwise difference of 12.78 nucleotides between any two genomes. The global nucleotide diversity (π) of the 2213 whole genomes was low (π = 0.00044 ± 0.00001). This diversity was not evenly distributed throughout the virus genome, with several high diversity peaks or hotspot mutations in ORF1ab, S gene and N gene being detected. N gene showed the highest peak of nucleotide diversity (π = 0.02934) (Figure 1). 

### 2.2. Spatial–Temporal Genetic Diversity of SARS-CoV-2

Over time, an increase in the global π values was observed, which coincided with the increase in COVID-19 cases from December 2019 to October 2020 (Figure 2). There was a slight decrease in π values in November 2020, but we only sampled until 13th November 2020, so a decrease for the month as a whole was expected. Regional analysis around the world showed that π values were low and similar to each other (United States of America (US) π = 0.00044 ± 0.00001, Latin America (LA) π = 0.00037 ± 0.00002, Europe (EU) π = 0.00043 ± 0.00002, Africa (AF) π = 0.00047 ± 0.00002, Asia (AS) π = 0.00042 ± 0.00001 and Oceania (OC) π = 0.00046 ± 0.00001), although AF and OC regions showed the highest diversities. 

Fluctuations in the π values with a tendency to increase over time were observed in US (January π = 0.00025 ± 0.00007–October π = 0.00071 ± 0.00003), EU (January π = 0.00008 ± 0.00002–November π = 0.00072 ± 0.00004), AF (February π = 0.00038 ± 0.00019–November π = 0.00100 ± 0.00004) and AS (December π = 0.00006 ± 0.00002–October π = 0.00062 ± 0.00005). In LA, there was an increase in the π values from March to August (π = 0.00028 ± 0.00002–π = 0.00046 ± 0.00004) but in September, a drastic decrease in the π value (π = 0.00025 ± 0.00012) was detected. While OC showed low diversity between February and September (π = 0.00014 ± 0.00006–π = 0.00020 ± 0.00002), the diversity increased dramatically (π = 0.00074 ± 0.00003 and π = 0.00080 ± 0.00022, respectively) during the months of October and November (Figure 3). 

### 2.3. Non-Synonymous Substitutions and Natural Selection

Among the 2213 whole genomes analyzed, we found 3178 polymorphic sites (*S*), of which a high proportion (58.5%, 1861 sites) were non-synonymous (*dN*) when compared with the reference strain, Wuhan-Hu-1. Although there was a large number of *dN* substitutions, the majority were neutral (*dN/dS* values were between −22.85 and 7.96 but not statistically significant). In general, it appears that the global population of SARS-CoV-2 is subject to purifying selection (*dN/dS* = −3.533; *p* < 0.01). 

When we analyzed *dN* substitutions in total, we identified seven in the global population of SARS-CoV-2 (Table 1) with frequencies > 10%. These seven frequencies varied by region: T85I and Q57H (nsp2) were the most frequent in US; I120F (nsp2) in OC; and R203K and G204R (N protein) in LA, AF and OC; P323L (nsp12) and D614G (S protein) were highly frequent in all regions. Positive selection was seen in T85S (*dN/dS* = 5.89; *p* < 0.01) and P323L (*dN/dS* = 7.49; *p* < 0.01), while I120F, D614G, Q57H and G204R had positive values of *dN/dS*, but these were not significant. Meanwhile, R203K presented a *dN/dS* negative value, but again, this was not significant (Table 1). 

### 2.4. Phylogeny and Dynamics of the Highly Frequent Global dN Substitutions

Phylogenetic analysis, using the Nexstrain nomenclature [14], showed that the 2213 genomes were grouped into seven clusters (Figure 4). G614 was related to clade 19A and the emergence of clade 20A, while L323 was related to the emergence of clade 20A. Clades 20B and 20C, and the subclades 20A.EU1 and 20A.EU2, arose from clade 20A. K203 and R204 were related to the emergence of clade 20B, while I85 was related to the emergence of clade 20C. H57 and F120 emerged into clades 20A and 20B, respectively. Finally, subclade 20A.EU1 was related to G614 and L323, and subclade 20A.EU2 to G614, L323 and H57 (Figure 4).

Subsequently, we performed a spatial–temporal analysis of the *dN* substitutions with the highest global frequencies (>75%), G614 and L323 (Table 1). G614 was detected for the first time in January 2020, and L323 in February 2020, with both substitutions presenting a high increase in their frequencies between February and March 2020. From April to September 2020, these substitutions were present in >90% of the isolates analyzed each month, and in October 2020, they presented in 100% of the isolates (Figure 5). By region, we observed fluctuations in their frequencies over time, but they were persistent in all regions. In the US and LA regions, both substitutions were detected from February 2020 to October 2020; in EU, G614 was detected from January 2020 to October 2020 and L323 from February 2020 to October 2020; in AF, both substitutions were detected from February 2020 to October 2020, while in AS and OC, they were detected from March 2020 to October 2020 (Figure 5). November was not included in the analysis because we only sampled until 13th November, and genomes could not be obtained from all regions. However, 43 isolates from EU, AF and OC were recovered in this month, and all of these presented the G614 and L323 substitutions.

Interestingly enough, we found that L323 and G614 showed similar frequencies and distributions, and both substitutions presented a strong linkage disequilibrium (LD) (R^2^ = 0.944; *p* < 0.001). The Nextrain phylogenetic tree indicated that these substitutions emerged early in the pandemic (G614, 2020-01-06 [IC 2019-12-27–2020-01-16]; L323, 2020-01-20, [IC 2019-01-11–2020-01-21]) and have spread all over the world (Appendix A).

### 2.5. Emergence and Transmission of New Variants of SARS-CoV-2

In addition to those previously described in Section 2.3 above, we investigated if there were *dN* substitutions with a significant increase in frequency by region. We found a *dN* substitution in the S gene (G22992A > S477N) with a *dN*/*dS* value of 1.92 (*p* = 0.485) and a frequency of 42.6% (*n* = 153) in the virus population from the OC region. An I120F substitution was also present in high frequency in OC (43.2%, *n* = 155) (Table 1).

The Nexstrain phylogenetic tree showed that F120 emerged in late March (2020-03-21; IC 2020-03-12–2020-03-27) in AS, and it spread in AS and OC regions (Appendix A). We detected F120 with moderate frequency (11.4%, *n* = 67) in AS (Bangladesh) from April 2020 to July 2020 and again in October 2020, and in a genome from EU (Wales) in September 2020. Meanwhile, N477 emerged in late May (2020-05-27; IC 2020-05-08–2020-06-05) in OC and it spread throughout this region (Appendix A and Figure 6A). 

F120 and N477 presented similar distributions and frequencies over time in OC, and were under strong linkage disequilibrium (R^2^ = 0.977; *p* < 0.001). Both substitutions were detected from June 2020 to October 2020, with their highest frequencies of 98–100% being seen in July 2020 to September 2020. However, between September 2020 and October 2020, there was a dramatic decrease in their frequencies from 100 to 32.4%, respectively (Figure 6B,C). 

A second cluster carrying the S477N substitution that included genomes from EU (France, Netherlands, Norway, Belgium and Denmark; *n* = 15, 78.95%), AF (Tunisia; *n* = 2, 10.52), AS (Hong Kong; *n* = 1, 5.26%) and OC (New Zealand; *n* = 1, 5.26%) (Figure 6A) was detected during September 2020 and November 2020. The phylogenetic trees showed that this cluster corresponded to subclade 20A.EU2 (Figure 4 and Figure 6A). Moreover, one genome from AF (Ivory Coast) located in clade 20A also carried this substitution. The Nexstrain phylogenetic tree suggests that subclade 20A.EU2 emerged in EU during July (2020-07-24; IC 2020-07-09–2020-08-03) (Appendix A).

### 2.6. Association between Amino acid Variation and Disease Severity

We focused on the *dN* substitutions located in the S protein (D614G), nsp12 (P323L) and N protein (R203K and G204R) to analyze associations between viral variants and disease severity. We found clinical information available for 118 patients; 21 (17.8%) patients from the low/mild disease group and 84 (71.2%) patients from the hospitalized/severe disease group had the L323 substitution. The G614 substitution was detected in 21 (17.8%) patients from the low/mild disease group and 81 (68.6%) from the hospitalized/severe disease group. K203 and R204 were detected in three (2.5%) patients from the low/mild disease group and 36 (30.5%) patients from the hospitalized/severe disease group. We found a significant association between the presence of G614 (*p* = 0.0047), L323 (*p* = 0.0005), and K203 and R204 (*p* = 0.0015) in patients with hospitalized/severe disease.

## 3. Discussion

Our results showed that the nucleotide diversity of the global population of SARS-CoV-2 has increased over time. Genome diversity was not homogeneous with regions showing high and low diversity. We found that more than 3000 mutations have emerged in the whole genome of the virus, and half of these have resulted in non-synonymous substitutions (*dN*), with most of them being neutral or likely neutral substitutions. The P323L and D614G substitutions in the global SARS-CoV-2 population have increased dramatically in their frequency over time. By October and November 2020, they were present in 100% of the virus population analyzed. Moreover, we detected two *dN* substitutions that spread in Oceania from July to October 2020, and we found a significant association between the G614, L323, K203 and R204 substitutions and hospitalized/severe disease. 

Analysis of the 2213 SARS-CoV-2 genomes revealed that they shared a high nucleotide identity, suggesting that the genetic variation is limited withing the global population of the virus. In the whole genome, we detected genomic regions of high and low nucleotide diversity, implying that some genomic regions are evolving faster than others [15,16]. This difference between genomic regions may be useful because regions with low diversity could be considered more suitable to develop and test new antiviral drugs, vaccines and detection methods (RT-PCR), in order to reduce the possibility of rapid drug resistance, immune system evasion and high numbers of false negatives when testing [17,18,19]. 

Global nucleotide diversity (π) varied over time and coincided with the increase in COVID-19 cases from December 2019 to October 2020. Previous studies have reported a positive association between sampling time and the evolution of the virus, indicating that more recent isolates have accumulated additional mutations more than older ones [15,19]. Although the number of samples per month in the current study was not homogeneous, the increase in diversity over time suggests that the global effective population size of SARS-CoV-2 is relatively high. Regionally, we also found that a tendency for diversity increased over time; however, there were fluctuations in the π values, which could be explained by the sample size per month per region and the over representation of a few genotypes in a given time. Infection patterns during outbreaks that might occur in a region over a determined time period could result in the over-representation of some mutations, resulting in a decrease in genetic diversity and a similar effect to that of natural selection [12,13]. 

Although we found a large number of *dN* substitutions, it is still unclear if they play a significant role since most of them are neutral or likely neutral. Seven of them presented frequencies > 10% in the global SARS-CoV-2 population and were detected in nsp2 (T85I and I120F), nsp12 (P323L), S protein (D614G), ORF3a (Q57H) and N protein (R203K and G204R). Additionally, we found a substitution in the S protein (S477N) with a high frequency in OC. Although the *dN/dS* values were positive for T85I, I120F, G614, L323, Q57H, G204R and S477N, only F120 and L323 presented statistical significance, indicating positive natural selection.

Interestingly, we found that G614 and L323, and F120 and N477, presented a strong LD, suggesting that this LD is the result of natural selection, and that the average fitness of isolates that carry both mutations could overcome the adequacy of each substitution [20], thereby suggesting that the LD among these two substitutions could persist over time [21]. However, more detailed bioinformatics and experimental analyses of LD, epistasis and natural selection are needed to understand the detailed evolution of these substitutions. 

Our results, together with those from the Nexstrain phylogenetic analysis, show that L323 and G614 emerged early in the pandemic in EU and AS, respectively, and these have spread worldwide with dramatic increases in frequency over time. Other substitutions were more frequent on a more regional basis, for example, F120 and N477, were highly frequent in OC. F120 emerged in AS and was then introduced to OC where it spread, while N477 emerged and spread in OC. The phylogenetic analysis showed that N477 has also been detected in other regions, principally in EU where it formed a well-defined clade (20A.EU2). In OC, N477 could be the result of an outbreak during the period June–October 2020 with cryptic transmission of SARS-CoV-2 in the region, which may be the case for other outbreaks that have been reported in US [22]. A recent study reported the presence of N477 in EU between June and September 2020 increasing its frequency over time, principally in France [23]. Our results indicate that the presence of this substitution in OC and EU was the result of two independent events (homoplasy), but the few cases observed in AF, AS and OC in clade 20A.EU2 suggest genetic flow from EU to those regions.

Another homoplasy event was the emergence of the mutation A23063T > N501Y in England and South Africa. In the middle of December 2020, a new outbreak in England of a new SARS-CoV-2 strain (named linage B.1.1.7) was reported. The more significant changes in this strain were the mutations A23063T and C23604A, which resulted in substitutions of N501Y and P681H, respectively, in the S protein. Although this strain was detected in late September 2020, a rapid increase in its frequency has since been reported in December 2020 in England, and it has spread to other countries from the UK, Europe, Africa, Asia, Oceania and America [24,25,26]. This variant is roughly 50–56% more transmissible than other virus variants but does not cause more severe disease [27,28]. Its rapid spread has been associated with the N501Y and P681H substitutions, which could be implicated in viral infectivity [29]. The variant (501Y.V2), which was first identified in South Africa in October 2020 [30], also carries the N501Y substitution. Recently, the British government reported two imported cases from South Africa [30]; those genomes had the N501Y substitution but did not share the same mutations in the B1.1.7 linage.

The combination of several mutations and phylogenetic associations provides information that helps to determine the origin of the viral genotypes, and so theoretically, if we know the origin of the genotypes, both local and imported cases can be detected allowing us to track the dynamics of viral spread at a local and global level. Thanks to molecular epidemiology, it has been possible to detect the emergence, introduction and transmission of new variants of the virus in different regions during this current pandemic [10,31,32,33,34]. This information is vital for developing public health interventions and policy to control viral spread.

Given its function, nsp12 is essential for the replication/transcription of the SARS-CoV-2 genome, and this protein serves as a target for the treatment of COVID-19. The P323L substitution is located in an interphase region of nsp12, and together with nsp7 and nsp8, it has been reported to play an important role in the formation of a protein complex [35], which provides structural stability to nsp12 for its processivity [36,37]. However, a previous report suggests that L323 could possess structural alterations [38] and an adverse effect on proofreading during the genomic replication of the virus [15]. Meanwhile, the P323L substitution is located in a pocket that has been predicted as a possible druggable site [39]; however, further research is needed to discover if the mutation could affect these properties. 

The S protein is a key factor for the entry of the virus to the host cell [40]. This protein has a receptor binding motif (RBM) that interacts with the ACE-2 receptor of the host cell [41]. The S447N substitution is located in the RBM and a recent study showed that S477 increases the affinity for the ACE-2 receptor [42]. Moreover, this substitution is part of an epitope recognized by human neutralizing antibodies [43], but further analysis is required to determine if N477 alters recognition by human antibodies. 

G614 has gained relevance since the presence of this substitution correlated with a higher capacity for infection by SARS-CoV-2 [44,45,46]. Moreover, studies in vitro have shown that this substitution is responsible for making the virus 2.4 times more infectious [47]. It has also been reported that the viral load in COVID-19 patients is higher than in those patients with isolates that do not present this substitution [48,49,50]. 

Furthermore, the S protein is among the elements targeted in the development of vaccines against SARS-CoV-2. Initial studies have shown that the presence of the D614G substitution produces a reduction in the neutralization titers using antibodies from convalescent plasma obtained from patients with COVID-19 [47]. This suggests that the substitution affected the antigenic response to the S protein. Recent reports have shown that mutations in the S protein are becoming more frequent as the pandemic spreads and that these mutations can have an increased capacity to spread [51,52]. To date, most serum samples from either volunteers in vaccines trials or patients recovering from COVID-19 have shown full or slightly diminished capacity to inactivate some of the more widespread SARS-CoV-2 variants, except for B.1.1.7 (N501Y substitution), 501Y.V2 (N501Y, K417N and E484K substitutions) and 501.V3 (N501Y and E484K substitutions), which have been able to cause a decrease in the neutralization assays using the aforementioned serum samples [53,54,55,56].

Although an effective COVID-19 vaccine could be the proximal solution to the SARS-CoV-2, genetic diversity in the S protein and its implication in host immune evasion must be taken into account in order to develop improved vaccines in the future, which may be required to protect against new mutations.

Finally, nsp2 is a helical transmembrane protein implicated in the modulation of the host cell environment [57], although its precise function remains unknow. Previous studies have reported that a stabilizing mutation in the endosome associate protein-like domain of the nsp2 could be associated with the more contagious phenotype of SARS-CoV-2 when compared with SARS-CoV [58]. The I120F substitution occurs in the N-terminal of nsp2, which is located in the extracellular region of the protein. We recommend the need for further study of its possible implications in virus fitness. 

Some genetic changes in SARS-Cov-2 may confer an evolutionary advantage, such as high transmissibility, evasion of the host immune system or future drug resistance, but they could also be implicated in clinical outcomes. We found that the D614G substitutions in the S protein, P323L in nsp12, and R203K and G204R in the N protein had a significant association with the disease severity. More specific studies will be needed to determine how these substitutions contribute to disease severity. Despite the present study including a small amount (5.3%) of COVID-19 clinical data form patients compared to the number of analyzed genomes, this could provide a preliminary approach for determining the association between SARS-CoV-2 and COVID-19 disease severity. Recently, Nagy et al. [59] showed a direct correlation between *dN* substitution and clinical outcomes. They found five *dN* substitutions in ORF8, nsp6, ORF3a, nsp4 and N protein related to mild disease, while 17 *dN* substitutions distributed in S protein, nsp12, ORF3a, N protein, nsp3, ORF6 and nsp7 were related to hospitalization and severe disease, including D614G, P323L, Q57H, R203K and G204R. Associations between the presence of the moderate and severe forms of COVID-19 in pediatric patients and P323L and D614G substitutions were also reported [60]. Moreover, P323L and D614G substitutions may correlate with higher fatality rates [61]. The development of a barcoding system could be useful to detect viral variants and diagnoses of severe COVID-19 disease.

One year after the emergence of the SARS-CoV-2, the virus continues to mutate, and it will keep accumulating novel mutations with possible clinical and therapeutic repercussions requiring the development of new strategies to reduce the burden of COVID-19 disease. Molecular epidemiologic surveillance needs to continue in order to detect genetic changes that might be involved in pathogenesis, host immune system evasion and/or future drug resistance, as well as its worldwide spread. Such information will contribute greatly to the development of more efficient interventions for SARS-CoV-2, as well as to providing a solid foundation for tackling other viral pandemics in the future.

## 4. Materials and Methods 

### 4.1. Sequences, Alignments and Quality Control

A total of 2500 complete genomes of SARS-CoV-2 from six regions around the world (United States of America (US), Latin America (LA), Europe (EU), Africa (AF), Asia (AS) and Oceania (OC)) were obtained randomly from NCBI [62] and GISAID [24] databases up to November 13, 2020. The genomes were aligned using MAFFT v7.3 [63] and revised by BioEdit v7.2 software [64], using the isolate from Wuhan, Hu-1 (GenBank: NC045512) as the reference strain. Non-coding regions were eliminated, as were all genomes that presented more than 15 non-determined (N) or other ambiguous nucleotides according to the IUPAC nucleotide code. For final analysis, we included 2213 genomes from 29,256 nucleotides distributed throughout the period between December 2019 and November 2020 (Appendix A).

### 4.2. Genetic Analyses

We used DnaSP v5.1 software [65] to determine the number of polymorphisms (*S*), nucleotide diversity (π), the number of non-synonymous (*dN*) substitutions and linkage disequilibrium (LD) given by the R^2^ index. The variations of π throughout the genome were estimated using a 50 bp window at 10 bp steps. To determine if diversity moves away from neutrality, the difference between synonymous and non-synonymous substitutions (*dN/dS*) was evaluated using the software MEGA v6.0 [66]. This estimation was based on the maximum joint likelihood of ancestral reconstruction states under the Muse–Gaut models [67] and Felsenstein’s codon substitution [68]. Moreover, the software calculates the probability of rejecting the null hypothesis of neutral evolution (*p* value). We obtained *dN* frequencies using *Jalview* v2.11 software [69].

### 4.3. Phylogenetic Analysis

A maximum likelihood phylogenetic tree of the 2213 SARS-CoV-2 genomes analyzed in this study on a background of 1888 reference genomes was constructed in Nextstrain [70] on 30 November 2020. In addition, we obtained a maximum likelihood phylogenetic tree (named Nexstrain phylogenetic tree) of SARS-CoV-2 obtained from Nextstrain [70] on the same date in order to localize the nucleotide changes and *dN* substitutions into their respective clades together with divergence times.

### 4.4. Clinical Classification and Genetic–Phenotype Association Analysis

For the 711 sequences downloaded from the GISAID database, we obtained the patient follow-up status (Appendix A). Only 118 sequences had informative patient status; the rest were marked as “unknown” or as “live”. The 118 patient samples with informative status were grouped into two categories: low/mild disease, which included patients who were marked as “asymptomatic”, “home”, “not hospitalized”, “outpatient”, “mild clinical signs without hospitalization” and “isolation”, and hospitalized/severe disease, which included patients who were marked as “hospitalized”, “released”, “deceased”, “intensive care unit” and “recovered”.

Association between genotype and disease severity was performed using a Fisher´s exact test and odds ratio calculation via a 2 × 2 contingency table, and statistical analysis was performed using Rstudio v3.2.2 [71].

## Figures and Tables

**Figure 1 pathogens-10-00184-f001:**
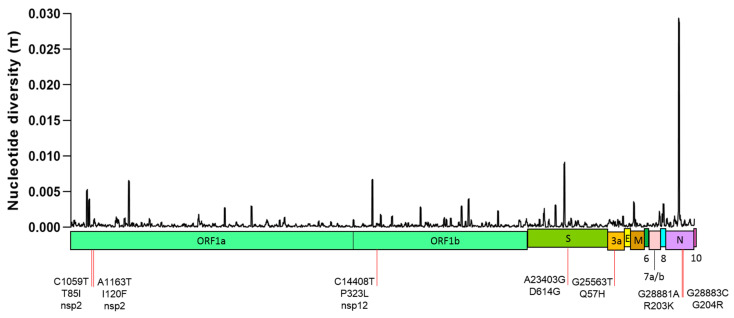
Nucleotide diversity (π) in a total of 2213 SARS-CoV-2 genomes. Several hotspot mutations were detected along the genome. Seven nucleotide substitutions with frequencies > 10% in the sample population are indicated, all of which resulted in amino acid non-synonymous (*dN*) substitutions. The π values were calculated within a sliding window of 50 bp moving with 10 bp steps.

**Figure 2 pathogens-10-00184-f002:**
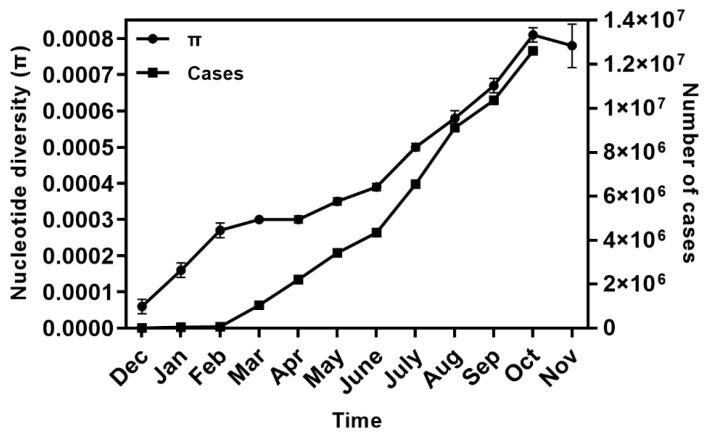
Temporal changes of SARS-Cov-2 nucleotide diversity (π) and monthly incidence of COVID-19 cases according to confirmed global cases from December 2019 to November 2020. The number of cases in November was not considered because we recorded until 13th November 2020 only. The number of isolates analyzed per month was as follows: Dec = 15, Jan = 103, Feb = 84, Mar = 628, Apr = 222, May = 221, June = 118, July = 233, Aug = 196 and Sep = 179, Oct = 171, Nov = 43. Abbreviations: Dec, December; Jan, January, Feb, February, Mar, March; Apr, April; Aug, August; Sep, September; Oct, October; Nov, November.

**Figure 3 pathogens-10-00184-f003:**
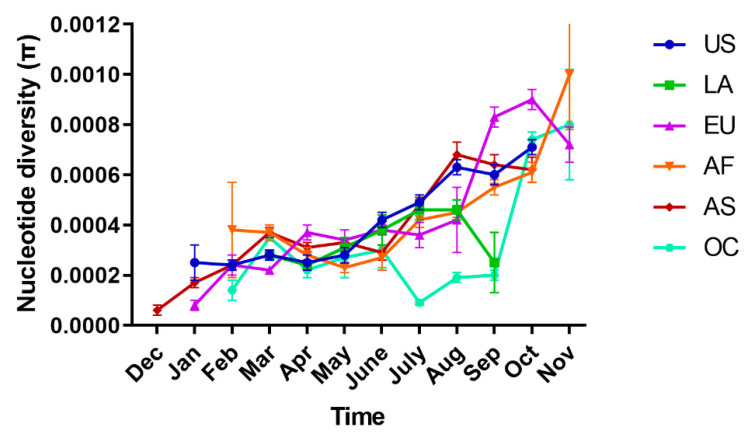
Temporal changes of SARS-Cov-2 nucleotide diversity (π) by region. Abbreviations: Dec, December; Jan, January, Feb, February, Mar, March; Apr, April; Aug, August; Sep, September; Oct, October; Nov, November. US, United States of America; LA, Latin America; EU, Europe; AF, Africa; AS, Asia; OC, Oceania.

**Figure 4 pathogens-10-00184-f004:**
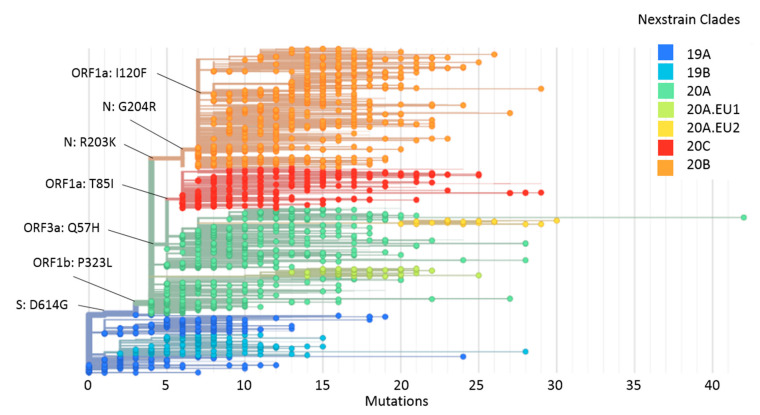
A Maximum-likelihood phylogeny of 2213 SARS-CoV-2 genomes. The branches with tip circles represent the 2213 genomes analyzed in the present work, and branches without a tip circle represent the 1888 reference genomes. The 7 *dN* substitutions analyzed here are located in the base of the nodes. The color of each circle indicates the clades to which it belongs according to Nexstrain nomenclature.

**Figure 5 pathogens-10-00184-f005:**
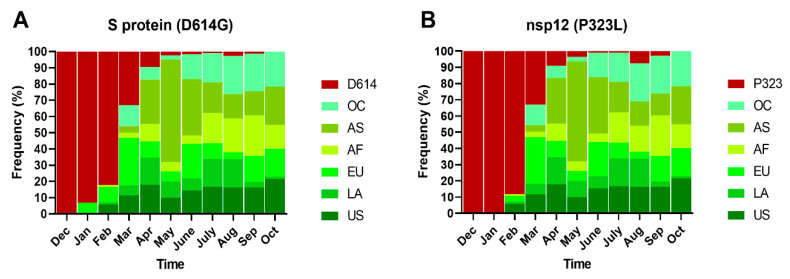
Spatial–temporal frequencies of D614G (**A**) and P323L (**B**) substitutions. Abbreviations: Dec, December; Jan, January, Feb, February, Mar, March; Apr, April; Aug, August; Sep, September; Oct, October. OC, Oceania; AS, Asia; AF, Africa; EU, Europe; LA, Latin America; US, United States of America. The number of isolates analyzed per month was as follows: Dec = 15, Jan = 103, Feb = 84, Mar = 628, Apr = 222, May = 221, June = 118, July = 233, Aug = 196, Sep = 179, and Oct = 171.

**Figure 6 pathogens-10-00184-f006:**
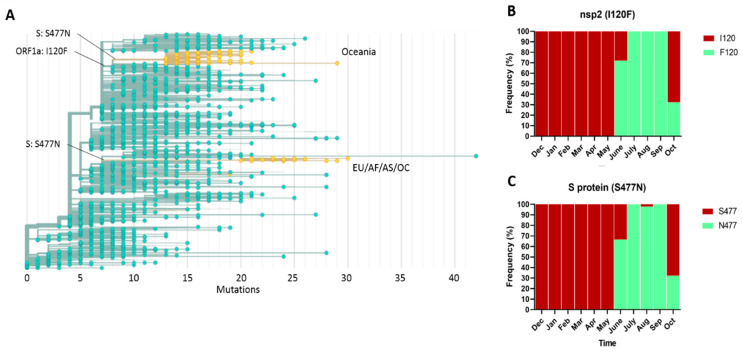
Maximum-likelihood phylogeny of 2213 SARS-CoV-2 genomes (**A**), and temporal frequencies of I120F (**B**) and S477N (**C**) substitutions in Oceania. F120 and N477 are located in the base of the nodes. The presence of N477 is indicated in yellow. Abbreviations: Dec, December; Jan, January, Feb, February, Mar, March; Apr, April; Aug, August; Sep, September; Oct, October. In Figures (**B**,**C**), the number of isolates analyzed per month was as follows: Jan = 1, Feb = 3, Mar = 144, Apr = 17, May = 6, June = 18, July = 42, Aug = 46, Sep = 42, and Oct = 37.

**Table 1 pathogens-10-00184-t001:** Non-synonymous Substitutions (*dN*) of Medium–High Frequency in the Global Population of SARS-CoV-2.

Nucleotide Change	Amino Acid Change	Genomic Location	*dN/dS* (*p* Value)	Distribution and Frequency (%)						
				US	LA	EU	AF	AS	OC	Global
C1059T	T85I	ORF1a (nsp2)	5.89 (0.009)	49.2	12.5	9.80	5.80	2.40	11.4	14.41
A1163T	I120F	ORF1a (nsp2)	4.79 (0.052)	0.00	0.00	0.20	0.00	11.4	43.2	10.08
C14408T	P323L	ORF1b (nsp12)	7.49 (0.002)	80.6	92.3	81.8	88.4	68.2	81.1	79.58
A23403G	D614G	S gene	2.42 (0.153)	80.3	90.9	85.3	92.6	69.0	81.1	80.80
G25563T	Q57H	ORF3	7.13 (0.105)	59.8	18.7	21.1	18.6	25.9	16.2	27.47
G28881A	R203K	N gene	−0.43 (0.805)	7.90	41.8	34.2	48.8	26.9	54.0	33.44
G28883C	G204R	N gene	1.79 (0.285)	7.90	41.8	34.0	48.3	26.7	54.0	33.30

Abbreviations: US, United States of America; LA, Latin America; EU, Europe; AF, Africa; AS, Asia; OC, Oceania.

## Data Availability

Not applicable.

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
