# Peer review of "Molecular Epidemiology Surveillance of SARS-CoV-2: Mutations and Genetic Diversity One Year after Emerging"

_pathogens, 2021, doi:10.3390/pathogens10020184_

Round 1

Reviewer 1 Report

Continuous molecular surveillance of SARS-CoV-2 will be necessary to detect and describe the transmission dynamics of new variants of the virus with clinical relevance. The authors of the mansucript performed genomic analysis to identify the changes in genetic diversity of SARS-CoV-2. This information is important to improve programs to control the virus.

Author Response

Dear Reviewer, please accept my sincere gratitude for your comments. 

Yours faithfully

Rosario Morales-Espinosa

Alejandro Flores-Alanis

Armando Cruz-Rangel

Flor Rodríguez-Gómez

James González

Carlos Alberto Torres-Guerrero

Gabriela Delgado

Alejandro Cravioto

Reviewer 2 Report

Dear Editor, 

the paper “Molecular epidemiology surveillance of SARS-CoV-2: mutations and genetic diversity one year after emerging” of Alejandro Flores-Alanis et al. reporting interesting genomic data about the changes in genetic diversity of SARS-CoV-2 between December 2019 and November 2020. In addition, through molecular surveillance, authors monitored the mutations that could be involved in viral fitness.  

The manuscript is well written and of great interest for the current SARS-CoV-2 pandemic. 

However, some minor revisions should be done before publication 

Abstract 

Authors should better specify the aim of the study in the abstract paragraph 

Results 

The comparison analysis among anamnestic data, mutations, and virus spread capability could improve the impact of manuscript 

Discussion 

Based on more recent data, authors could discuss with more details the possible effect of reported mutations in the efficacy of the currently approved vaccine 

Author Response

Dear Reviewer, please accept my sincere gratitude for your comments. I know how busy you are, so I really appreciate the time you spent in reviewing this new version of the manuscript, which now has all the changes that you requested.

Reviewer 2,

The paper “Molecular epidemiology surveillance of SARS-CoV-2: mutations and genetic diversity one year after emerging” by Alejandro Flores-Alanis et al. provides important genomic data about the changes in genetic diversity of SARS-CoV-2 between December 2019 and November 2020. In addition, through molecular surveillance, the authors monitored the mutations that could be involved in viral fitness. 

The manuscript is well written and of great interest for the current SARS-CoV-2 pandemic. 

However, some minor revisions should be done before publication 

Abstract 

Authors should better specify the aim of the study in the abstract paragraph

We reworded the manuscript in lines 18-20.

Results 

The comparison analysis among anamnestic data, mutations, and virus spread capability could improve the impact of manuscript

We added an association analysis between the virus genotypes and disease severity (mild disease and hospitalized/severe disease) using a Fisher´s exact test and odds ratio calculation using a 2x2 contingency table. We found that out of the 118 patients, 21 (17.8%) patients from the low/mild-disease group and 84 (71.2%) patients from the hospitalized/severe-disease group showed the L323 substitution. The G614 substitution was detected in 21 (17.8%) subjects from the low/mild-disease group and 81 (68.6%) from the hospitalized/severe-disease group. While K203 and R204 substitutions were detected in 3 (2.5%) patients from the low/mild-disease group and 36 (30.5%) patients from the hospitalized/severe-disease group. We found a significant association between the presence of G614 (p=0.0047), L323 (p=0.0005), and K203 and R204 (p=0.0015) with hospitalized/severe disease (this information is in results section, line 212-222). Unfortunately, of the 2,213 complete genomes analyzed in this study, we could only recover anamnestic data from 118 isolates; the rest of the isolates (1950) did not contain this information. This information has been included in Material and Methods, lines 408-419, and in the discussion, lines 351-368.  

Discussion 

Based on more recent data, authors could discuss with more details the possible effect of reported mutations in the efficacy of the currently approved vaccine

A more detailed discussion as suggested has been included in lines 327-343.

Finally, the English language was revised again.

Reviewer 3 Report

I assessed the manuscript by Flores-Alanis et al which aimed to assess the molecular epidemiology surveillance of SARS-CoV-2. The authors used an international repository of viral sequences (GISAID) and analysed 2500 complete genomes of SARS-CoV-2 from 6 continents. Overall the manuscript is well written and the of interest for physicians involved in the care of COVID-19 patients. In particular, the topic is actual considering the emergence of a new variant in England which is challenging the capacity of the UK national health system. Nevertheless, some concerns regard:

  • The limited sample of viral sequences
  • The novelty of the findings when considering what is already known
  • The added value of the analysis conducted regarding the knowledge of the pandemic and the potential new implication driven by the present research

Author Response

Dear Reviewer, please accept my sincere gratitude for your comments. I know how busy you are, so I really appreciate the time you spent in reviewing this new version of the manuscript, which now has all the changes that you requested.

Reviewer 3.

I assessed the manuscript by Flores-Alanis et al which aimed to assess the molecular epidemiology surveillance of SARS-CoV-2. The authors used an international repository of viral sequences (GISAID) and analysed 2500 complete genomes of SARS-CoV-2 from 6 continents. Overall the manuscript is well written and the of interest for physicians involved in the care of COVID-19 patients. In particular, the topic is actual considering the emergence of a new variant in England which is challenging the capacity of the UK national health system. Nevertheless, some concerns regard:

  • The limited sample of viral sequences
  • The annotation of complete genomes of the COVID-19 virus is constant and due to the great global impact of the pandemic, the number of genomes deposited in the database increases exponentially, however, we believe that this number of genomes is representative of the genetic diversity that is occurring in the virus. Although the number of samples is small in relation to what is deposited in the database, our study shows the behavior of the virus over a year.
  •  
  • The novelty of the findings when considering what is already known
  • The added value of the analysis conducted regarding the knowledge of the pandemic and the potential new implication driven by the present research
  • We added an association analysis between the virus genotypes and disease severity (mild disease and hospitalized/severe disease) using a Fisher´s exact test and odds ratio calculation using a 2x2 contingency table. We found that out of the 118 patients, 21 (17.8%) patients from the low/mild-disease group and 84 (71.2%) patients from the hospitalized/severe-disease group showed the L323 substitution. The G614 substitution was detected in 21 (17.8%) subjects from the low/mild-disease group and 81 (68.6%) from the hospitalized/severe-disease group. While K203 and R204 substitutions were detected in 3 (2.5%) patients from the low/mild-disease group and 36 (30.5%) patients from the hospitalized/severe-disease group. We found a significant association between the presence of G614 (p=0.0047), L323 (p=0.0005), and K203 and R204 (p=0.0015) with hospitalized/severe disease (this information is in results section, line 209-220). Unfortunately, of the 2,213 complete genomes analyzed in this study, we could only recover anamnestic data from 118 isolates; the rest of the isolates (1950) did not contain this information. This information has been included in Material and Methods, lines 408-419, and in the discussion, lines 352-368.  

Finally, the English language was revised again.